# Rethinking Radical Surgery in Interval Debulking Surgery for Advanced-Stage Ovarian Cancer Patients Undergoing Neoadjuvant Chemotherapy

**DOI:** 10.3390/jcm9041235

**Published:** 2020-04-24

**Authors:** Yong Jae Lee, Jung-Yun Lee, Eun Ji Nam, Sang Wun Kim, Sunghoon Kim, Young Tae Kim

**Affiliations:** Department of Obstetrics and Gynecology, Institute of Women’s Life Medical Science, Yonsei University College of Medicine, Seoul 03722, Korea; svass@yuhs.ac (Y.J.L.); NAHMEJ6@yuhs.ac (E.J.N.); SAN1@yuhs.ac (S.W.K.); SHKIM70@yuhs.ac (S.K.); YTKCHOI@yuhs.ac (Y.T.K.)

**Keywords:** ovarian cancer, radical surgery, disease burden, residual disease, neoadjuvant chemotherapy

## Abstract

The aim of this study is to evaluate the effects on survival outcomes of the disease burden before interval debulking surgery (IDS), surgical complexity, and residual disease after IDS in advanced-stage ovarian cancer treated with neoadjuvant chemotherapy (NAC). We reviewed the data of 268 epithelial ovarian cancer patients who had received three or four cycles of NAC and undergone optimal resections through IDS. The Kaplan–Meier method and Cox regression analysis were used to assess the effects of disease burden (peritoneal cancer index (PCI)), degree of complexity of surgery (surgical complexity score/s (SCS)), and extent of residual disease. In no residual disease (R0) patients, those with intermediate/high SCS had shorter progression-free survival (PFS; *p* = 0.001) and overall survival (OS; *p* = 0.001) than patients with low SCS. An analysis of a subset of patients with R0 and low PCIs showed those with intermediate/high SCS had worse PFS and OS than patients with low SCS (*p* = 0.049) and OS (*p* = 0.037). In multivariate analysis, patients with R0 as a result of intermediate/high SCS fared worse than patients whose R0 was achieved by low SCS (PFS hazard ratio (HR) 1.80, 95% CI 1.05–3.10; OS HR 5.59, 95% CI 1.70–18.39). High PCIs at the time of IDS, high SCS, and residual disease signal poor prognoses for patients treated with NAC.

## 1. Introduction

Ovarian cancer is a highly lethal gynecologic malignancy worldwide and usually is diagnosed at only an advanced stage [1]. Maximal debulking surgery plus adjuvant platinum-based chemotherapy has become the accepted primary treatment for the disease in its advanced stages. Optimal cytoreduction often requires radical procedures to achieve “no gross residual” disease at the time of surgery. In patients who underwent primary debulking surgery (PDS), the absence of residual disease after cytoreductive surgery is a strong predictor of improved survival [2,3]. Radical procedures to remove all visible disease before administration of chemotherapy has become the norm in many centers [4,5]. However, several studies have shown that patients with an initial high disease burden will nevertheless have a worse prognosis despite optimal resection with aggressive surgery [6,7]. These results suggest that tumor biology is more important than surgical cytoreduction.

Neoadjuvant chemotherapy (NAC) followed by interval debulking surgery (IDS) was introduced to diminish the initial disease burden and increase the likelihood of successful IDS [8,9,10,11]. With NAC becoming more common as an alternative treatment, the debate over the role of radical surgery in IDS is growing. To date, few studies have assessed the effects of initial disease burden—the disease burden before IDS—and surgical complexity in advanced-stage ovarian cancer treated with NAC [12]. In patients treated with NAC, optimal cytoreductive surgery and no residual disease (R0) at the time of IDS resulted in the most favorable survival outcomes [13,14]. Patients who had achieved R0 at the time of IDS often had less aggressive tumor biology and responded favorably to NAC. However, the debate continues on whether aggressive surgery during IDS to achieve R0 may overcome unfavorable tumor biology and achieve outcomes comparable to those of patients who achieve R0 with less aggressive surgery and respond well to NAC. 

The aim of this study is to evaluate the effects on survival outcomes of the disease burden before IDS, the surgical complexity of IDS, and the presence of residual disease after IDS. We hypothesize that the patients who achieved R0 with radical surgery for a high disease burden at the time of IDS might have survival rates similar to those who had R0 with less aggressive surgery for a low disease burden at the time of IDS.

## 2. Materials and Methods

### 2.1. Study Populations

We retrospectively reviewed the medical records of 313 patients with pathologically confirmed ovarian cancer who received NAC from 2006 to 2018 at the Yonsei Cancer Center, Seoul, South Korea. Inclusion criteria were as follows: (1) histopathologically confirmed FIGO Stage III or Stage IV ovarian, fallopian tube, or primary peritoneal carcinoma, (2) patients who underwent IDS after NAC, and (3) patients who received three or four cycles of NAC before undergoing IDS. We excluded patients who had suboptimal surgery at the time of IDS (*n* = 29) and patients who did not undergo IDS after NAC (*n* = 16). After this review, 268 patients met our criteria. We categorized them according to their disease burden at the time of IDS, the complexity of their surgery, and the presence of residual disease after undergoing IDS (Figure 1).

### 2.2. Treatment 

All patients received taxane and platinum combination chemotherapy. Other treatments, such as radiation or endocrine therapy, were not performed before surgery. At the time of IDS, the degree of disease burden was determined according to the peritoneal cancer index (PCI), as described by Harmon and Sugarbaker [15]. Different cutoff points were tested by receiver-operating characteristic (ROC) curves to determine the best cutoff point within the PCI to predict survival. 

All patients underwent surgery with the intent to remove all visible and palpable tumors. The complexity of the procedures used during IDS was classified, in accordance with previously published protocols, as low (surgical complexity score/s (SCS) 1 to 3), intermediate (SCS 4 to 7), or high (SCS ≥ 8) [16]. Each procedure was assigned a score from 1 to 3 to designate its complexity. The procedures designated as 1 were hysterectomy, bilateral salpingo-oophorectomy, omentectomy, pelvic lymphadenectomy, para-aortic lymphadenectomy, abdominal peritoneum strippings, and small bowel resections. Designated as 2 in complexity were large bowel resections, diaphragm stripping/resections, splenectomy, and liver resections. Recto-sigmoidectomy with anastomosis was designated as 3, the most complex. 

In residual tumor designations, R0 is defined as the removal of all tumors, eliminating all residual disease. Minimal residual disease (MR) is defined as incomplete removal of all tumors, leaving microscopic or macroscopic residual disease smaller than 1 cm. The present study was reviewed and approved by our Institutional Review Board on 18 May 2018 (IRB number: 4-2018-0518).

### 2.3. Statistical Analysis

Descriptive data are reported as the median (range) or frequency (percentage). Categorical variables were compared with the chi-square or Mann–Whitney U-test. Responses were assessed according to the Response Evaluation Criteria in Solid Tumors criteria, version 1.1. We defined progression-free survival (PFS) as the time from the date of diagnosis to disease progression; overall survival (OS) was measured from the date of diagnosis to death or to the date of the last follow-up. Survival analysis was performed using the Kaplan–Meier method with a log-rank test. Cox regression analysis was used to evaluate the effects of the prognostic factors, expressed as hazard ratios (HR) with 95% confidence intervals (CI). In all analyses, *p* < 0.05 was considered statistically significant. The statistical analyses were performed with the SPSS statistical software (version 21.0; IBM Corp., Armonk, NY, USA).

## 3. Results

### 3.1. Patients’ Characteristics

The ROC curve analysis determined a high disease burden of PCI 6 as the optimal cutoff (Appendix A). The clinical characteristics of patients based on disease burden and SCS are shown in Table 1. Patients were categorized according to low (≤6, *n* = 123) and high (>6, *n* = 145) PCIs based on disease burden at the time of IDS. The high PCI group had a higher SCS than the low group (42.8% vs. 10.6%) but a lower R0 rate (33.1% vs. 61.0%). Patients were also stratified according to low (*n* = 52), intermediate (*n* = 141), and high (*n* = 75) scores in surgical complexity. In patients with a high SCS, the incidence of high PCIs was higher than in those patients with low SCS (82.7% vs. 9.6%), but the R0 rate was lower (38.7% vs. 69.2%). Patients with a high PCI tended to have Stage IV disease (*p* = 0.007), MR (*p* < 0.001), and higher SCS (*p* < 0.001). Patients with higher SCS tended to have Stage IV disease (*p* = 0.007), non-high-grade serous tumors (*p* = 0.006), MR (*p* = 0.001), and a high PCI (*p* < 0.001). 

### 3.2. Effects of Disease Burden, SCS, and Residual Disease on Survival 

Patients with a high PCI at the time of IDS had a lower PFS (median, 15.1 vs. 24.6 months; *p* < 0.001) and OS (median, 45.1 vs. 76.5 months; *p* < 0.001), respectively (Figure 2A,B). Patients with MR had worse PFS (median, 18.2 vs. 22.0 months; *p* = 0.001) and OS (median, 51.8 vs. 79.2 months; *p* = 0.007) than patients with R0, respectively (Figure 2C,D). Patients with intermediate/high SCS had worse PFS (median, 17.2 vs. 26.8 months; *p* < 0.001) and OS (median, 49.7 vs. 89.2 months; *p* = 0.001), respectively, than patients with low SCS (Figure 2E,F). Low PCIs in patients with R0 were significantly associated with improved PFS (median, 26.8 vs. 15.3 months; *p* < 0.001) and OS (median, not reached vs. 63.4 months; *p* < 0.001) (Appendix A). Low PCIs in patients with MR were associated with improved PFS (median, 20.5 vs. 14.7 months; *p* = 0.002) and OS (median, 51.8 vs. 42.5 months; *p* = 0.045) (Appendix A). 

### 3.3. SCS on Survival

Kaplan–Meier curves for PFS and OS as stratified by residual disease and SCS (R0 with low SCS, R0 with intermediate/high SCS, MR with low SCS, and MR with intermediate/high SCS) are shown in Figure 3. Patients who achieved R0 through a low SCS had better PFS (median, 31.5 vs. 19.9 months; *p* = 0.049) and OS (median, not reached vs. 67.6 months; *p* = 0.022) than those patients whose R0 was achieved with an intermediate/high SCS. Among the MR patients, their PFS (median, 20.3 vs. 16.3 months; *p* = 0.096) and OS (median, 49.9 vs. 46.6 months; *p* = 0.848) did not differ significantly from those patients with low and intermediate/high SCS.

A subanalysis of patients with R0 achieved by low SCS with low PCIs had better PFS (median, 43.8 vs. 26.6 months; *p* = 0.049) and OS (median, not reached vs. 67.6 months; *p* = 0.037) than patients with intermediate/high SCS (Figure 4). Patients with MR and low PCIs did not differ significantly in PFS (median, 22.2 vs. 20.5 months; *p* = 0.303) and OS (median, 66.1 vs. 51.8 months; *p* = 0.921) from patients with low and intermediate/high PCIs (Appendix A). Among patients with high PCIs, those who achieved R0 through intermediate/high SCS fared no better in PFS (median, not reached vs. 17.0 months; *p* = 0.683) and OS (median, not reached vs. 63.4 months; *p* = 0.884) than those with low SCS (Appendix A). Surgical complexity had no bearing on either PFS (median, 12.4 vs. 14.7 months; *p* = 0.963) or OS (median, 34.7 vs. 42.5 months; *p* = 0.333) for patients with high PCIs and MR. 

Table 2 shows the results of multivariate Cox regression analyses of PFS and OS. The intermediate/high SCS (HR, 1.80; 95% CI 1.05–3.10) of RO patients, the low SCS (HR 2.25; 95% CI 1.07–4.74) of MR patients, and the intermediate/high SCS (HR 2.94; 95% CI 1.74–4.97) of MR patients were all independent prognostic factors associated with a higher risk of cancer progression. The OS results were similar; the intermediate/high SCS (HR 5.59; 95% CI 1.70–18.39) of RO patients, the low SCS (HR 9.92; 95% CI 2.67–36.90) of MR patients, and the intermediate/high SCS (HR 8.91; 95% CI 2.78–28.63) of MR patients were all independent prognostic factors associated with a higher risk of death.

## 4. Discussion

In our study, we evaluated the relationships between disease burden at the time of IDS, SCS, and residual disease after IDS on survival outcomes in advanced-stage ovarian cancer patients treated with NAC. We showed that a high PCI at the time of IDS intermediate/high SCS resulted in poor outcomes compared with those of patients with low PCIs and low SCS. Patients with high PCIs underwent more radical procedures in IDS than those patients with low PCIs. However, the rate of R0 was low in patients with high PCIs. Furthermore, patients with high PCIs who achieved R0 through more radical procedures gained no survival benefits. Consistent with previous studies, our study showed significant survival benefits for patients with R0 over those with MR. Therefore, aggressive surgery may be warranted if R0 can be achieved. However, R0 achieved this way did not translate into differential effects on survival outcomes in patients with high PCIs; however, R0 achieved with less aggressive procedures at the time of IDS had better survival outcomes in patients with low PCIs. Our study suggests that tumor biology is the most important prognostic factor in advanced-stage ovarian cancer patients treated with NAC.

The importance of maximal cytoreduction to eliminate residual disease has become widely accepted in the primary treatment of ovarian cancer patients [17]. Several studies have suggested that aggressive surgery can overcome tumor biology and that patients undergoing radical procedures can achieve improved survival outcomes [18,19,20]. In contrast, some investigators have demonstrated that the distribution of cancers with poor prognoses may be determined through their tumor biology [6,7,12]. Horowitz et al. [7] explored the relationship between preoperative disease burden and surgical complexity in advanced-stage ovarian cancer patients treated with PDS. In patients who achieved R0 through PDS, those patients with low or moderate preoperative disease burden had better survival outcomes than those with high preoperative disease burden. Such outcomes suggest that tumor biology, not residual disease, is the greater determinant of survival. Crawford et al. [6] reported similar findings in the SCOTROC-1 trial. In the NAC settings, Davidson et al. [12] reported a relationship between more aggressive surgery and worse survival outcomes in advanced-stage ovarian cancer patients treated with NAC. They also suggested that tumor biology might be the most significant prognostic factor. The degree to which aggressive surgery can overcome the negative impact of disease burden after NAC remains unclear. 

Our study showed that the patients who achieved R0 through aggressive surgery with high disease burden at the time of IDS had a poorer prognosis than those who achieved R0 with less aggressive surgery in conjunction with a low disease burden. Therefore, an additional therapeutic strategy is needed to reduce the disease burden after NAC, such as adding bevacizumab or immune checkpoint inhibitors to conventional chemotherapy. Rouzier et al. [21] reported that patients treated with NAC with bevacizumab achieved a higher complete resection rate through IDS than NAC without bevacizumab. Böhm et al. [22] suggested that NAC may enhance patients’ immune responses, opening a window of opportunity for immune checkpoint inhibitors for patients with advanced-stage ovarian cancer. Based on these results, we began an ongoing clinical trial examining the benefit of combination therapy with conventional chemotherapy, anti-PD-L1, and anti-CTLA-4 to improve the chemotherapy response in NAC [23]. Patients with a low disease burden after NAC who have had a good response to chemotherapy may consider minimally invasive surgery at the time of IDS. Several studies have shown that the incorporation of minimally invasive surgery with IDS is safe and feasible with acceptable R0 rates without perioperative complications [12,24]. In addition, in our study, patients with optimal resections at IDS had worse survival than patients with complete resections; this indicates additional adjuvant therapies based on genomic profiling of residual tumors after NAC may merit consideration. 

Our study has numerous limitations. First, the number of patients was small, and the data is from a single center. In particular, the number of patients receiving low SCS but high PCIs before IDS and those receiving high SCS high with low PCIs before IDS were relatively small. Second, NAC has been used as an alternative treatment in our institution since late 2010; thus, our cohort was limited by a short period of follow-up. Third, analysis of various factors, such as sarcopenia or body composition, which can affect survival outcomes in patients undergoing NAC is needed [25,26].

In conclusion, this study showed that aggressive surgery at the time of IDS does not translate into a survival benefit even in patients who achieved R0 after IDS. The disease burden left after NAC remained a significant prognostic factor. Therefore, the strategy to improve chemotherapy response before IDS might be a more rational approach to improve outcomes.

## Figures and Tables

**Figure 1 jcm-09-01235-f001:**
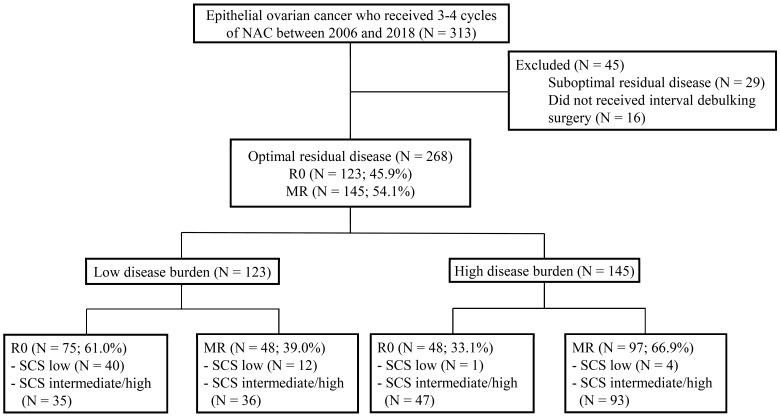
Flow diagram of the study population. NAC, neoadjuvant chemotherapy; R0, no residual disease; MR, <1 cm of residual disease; SCS, surgical complexity score.

**Figure 2 jcm-09-01235-f002:**
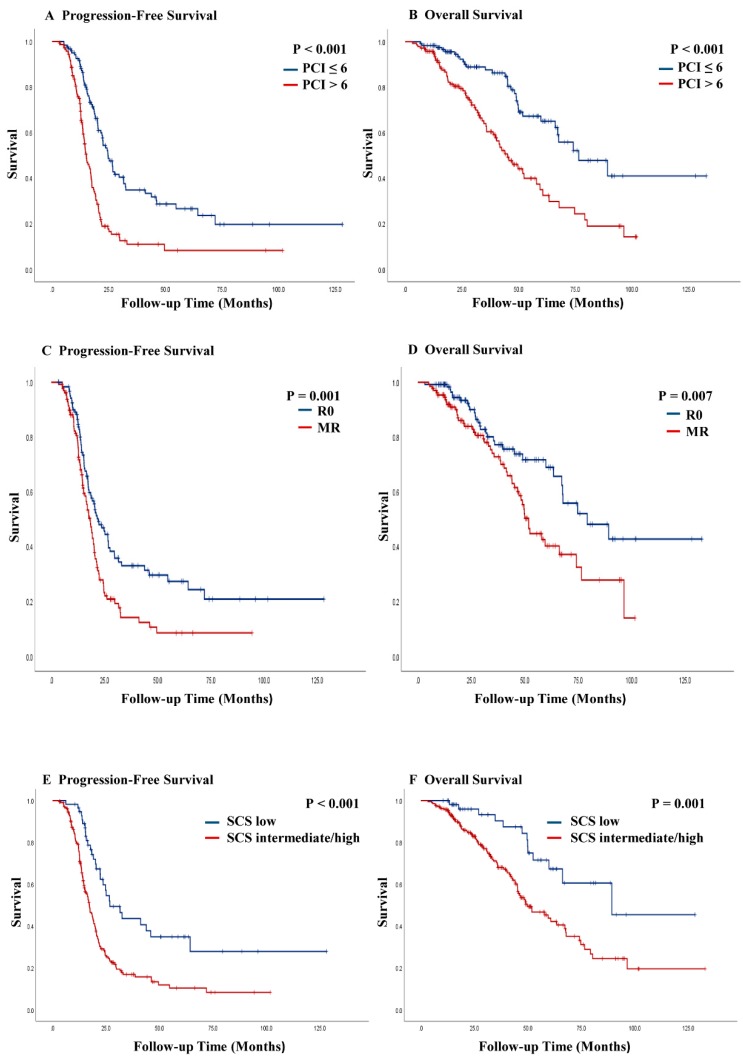
Kaplan–Meier curves of progression-free survival and overall survival stratified by disease burden after neoadjuvant chemotherapy (**A**,**B**), residual disease (**C**,**D**), and surgical complexity score (**E**,**F**). PCI, peritoneal cancer index; R0, no residual disease; MR, <1 cm of residual disease; SCS, surgical complexity score.

**Figure 3 jcm-09-01235-f003:**
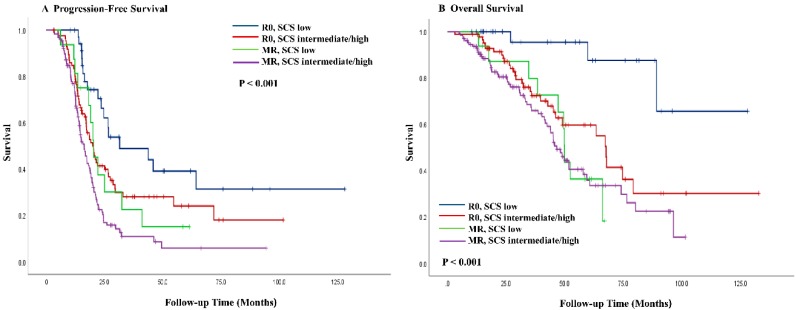
Kaplan–Meier curves of progression-free (**A**) survival and overall survival (**B**) stratified by residual disease and surgical complexity score. R0, no residual disease; MR, <1 cm of residual disease; SCS, surgical complexity score.

**Figure 4 jcm-09-01235-f004:**
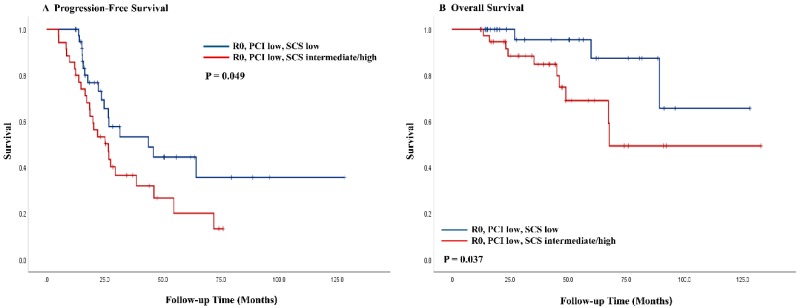
Kaplan–Meier curves of progression-free (**A**) survival and overall survival (**B**) stratified by surgical complexity score in no residual disease patients with low disease burden before interval debulking surgery. R0, no residual disease; MR, <1 cm of residual disease; PCI, peritoneal cancer index; SCS, surgical complexity score.

**Table 1 jcm-09-01235-t001:** Patient characteristics by disease burden after neoadjuvant chemotherapy (NAC) and surgical complexity score (SCS).

	Disease Burden			SCS			
	PCI ≤ 6 (*n* = 123)	PCI > 6 (*n* = 145)	*p*	Low (*n* = 52)	Intermediate (*n* = 141)	High (*n* = 75)	*p*
Age, median (range), years	58 (31–80)	58 (31–78)	0.624	59 (38–76)	58 (31–80)	56 (31–78)	0.917
ASA score, *n* (%)			0.585				0.808
1	21 (17.1%)	21 (14.5%)		10 (19.2%)	24 (17.0%)	8 (10.7%)	
2	64 (52.0%)	68 (46.9%)		26 (50.0%)	70 (49.7%)	36 (48.0%)	
3	38 (30.9%)	54 (37.2%)		16 (30.8%)	45 (31.9%)	31 (41.3%)	
4	0 (0%)	1 (0.7%)		0 (0%)	1 (0.7%)	0 (0%)	
Not available	0 (0%)	1 (0.7%)		0 (0%)	1 (0.7%)	0 (0%)	
CA-125 level, median (range), U/mL	1474.1 (44.3–30000.0)	1999.5 (70.9–30000.0)	0.066	1433.2 (44.3–30000.0)	1758.0 (60.1–30000.0)	1974.9 (75.2–20685.7)	0.415
FIGO stage, *n* (%)			0.007				0.007
III	68 (55.3%)	56 (38.6%)		32 (61.5%)	67 (47.5%)	25 (33.3%)	
IV	55 (44.7%)	89 (61.4%)		20 (38.5%)	74 (52.5%)	50 (66.7%)	
Histologic type, *n* (%)			0.170				0.006
HGSC	117 (95.1%)	135 (93.1%)		51 (98.1%)	134 (95.0%)	67 (89.4%)	
Endometrioid	2 (1.6%)	0 (0%)		1 (1.9%)	1 (0.7%)	0 (0%)	
Mucinous	0 (0%)	2 (1.4%)		0 (0%)	1 (0.7%)	1 (1.3%)	
Clear cell	1 (0.8%)	5 (3.4%)		0 (0%)	0 (0%)	6 (8.0%)	
Others	3 (2.5%)	3 (2.1%)		0 (0%)	5 (3.6%)	1 (1.3%)	
Grading			0.172				0.092
1	2 (1.6%)	3 (2.1%)		0 (0%)	5 (3.5%)	0 (0%)	
2	12 (9.8%)	18 (12.4%)		5 (9.6%)	14 (9.9%)	11 (14.7%)	
3	94 (76.4%)	117 (80.7%)		39 (75.0%)	111 (78.7%)	61 (81.3%)	
Not available	15 (12.2%)	7 (4.8%)		8 (15.4%)	11 (7.8%)	3 (4.0%)	
Residual disease, *n* (%)			<0.001				0.001
0	75 (61.0%)	48 (33.1%)		36 (69.2%)	58 (41.1%)	29 (38.7%)	
<1 cm	48 (39.0%)	97 (66.9%)		16 (30.8%)	83 (58.9%)	46 (61.3%)	
SCS			<0.001				
Low (≤3)	47 (38.2%)	5 (3.4%)		-	-	-	
Intermediate (4–7)	63 (51.2%)	78 (53.8%)		-	-	-	
High (≥8)	13 (10.6%)	62 (42.8%)		-	-	-	
Disease burden							<0.001
PCI ≤ 6	-	-		47 (90.4%)	63 (44.7%)	13 (17.3%)	
PCI > 6	-	-		5 (9.6%)	78 (55.3%)	62 (82.7%)	

ASA, American Society of Anesthesiologists; FIGO, International Federation of Gynecology and Obstetrics; HGSC, high-grade serous carcinoma; NAC, neoadjuvant chemotherapy; PCI, peritoneal cancer index; SCS, surgical complexity score; R0, no residual disease; R1, residual disease less than 1 cm.

**Table 2 jcm-09-01235-t002:** Univariate and multivariate analyses for progression-free and overall survival using a Cox proportional hazards model.

Variables	PFS				OS			
	Univariate Analysis		Multivariate Analysis		Univariate Analysis		Multivariate Analysis	
	HR (95% CI)	*p*	HR (95% CI)	*p*	HR (95% CI)	*p*	HR (95% CI)	*p*
Age, years								
≤58	1		1		1		1	
>58	1.05 (0.78–1.41)	0.736	1.04 (0.75–1.45)	0.810	1.09 (0.74–1.62)	0.656	1.14 (0.75–1.74)	0.536
ASA score								
1–2	1		1		1		1	
3–4	3.14 (0.43–22.66)	0.257	3.04 (0.41–22.57)	0.276	2.38 (0.33–17.26)	0.391	1.69 (1.07–2.68)	0.026
FIGO stage								
III	1		1		1		1	
IV	1.84 (1.36–2.49)	<0.001	1.70 (1.25–2.32)	0.001	1.32 (0.88–1.97)	0.177	1.18 (0.77–1.81)	0.444
Histology								
HGSC	1		1		1		1	
Non-HGSC	0.75 (0.33–1.71)	0.497	0.76 (0.33–1.72)	0.759	1.95 (0.90–4.21)	0.089	2.09 (0.95–4.64)	0.069
Residual disease x SCS								
R0, SCS low	1		1		1		1	
R0, SCS intermediate/high	1.44 (0.88–2.34)	0.144	1.80 (1.05–3.10)	0.034	2.00 (0.93–4.33)	0.075	5.59 (1.70–18.39)	0.005
MR, SCS low	3.39 (1.17–9.80)	0.024	2.25 (1.07–4.74)	0.034	7.15 (2.19–23.38)	0.001	9.92 (2.67–36.90)	0.001
MR, SCS intermediate/high	3.02 (1.94–4.70)	<0.001	2.94 (1.74–4.97)	<0.001	3.98 (1.97–8.03)	<0.001	8.91 (2.78–28.63)	<0.001

ASA, American Society of Anesthesiologists; CI, confidence interval; ASA, American Society of Anesthesiologists; CI, confidence interval; FIGO, International Federation of Gynecology and Obstetrics; HGSC, high-grade serous carcinoma; HR, hazard ratio; OS, overall survival; PCI, peritoneal cancer index; PFS, progression-free survival; R0, no residual disease; MR, <1 cm residual disease; SCS, surgical complexity score.

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
