# Peer review of "Rethinking Radical Surgery in Interval Debulking Surgery for Advanced-Stage Ovarian Cancer Patients Undergoing Neoadjuvant Chemotherapy"

_jcm, 2020, doi:10.3390/jcm9041235_

Round 1

Reviewer 1 Report

The authors well demonstrated both of disease burden and complexity of surgery are closely related to patient survival and tumor recurrence after interval debulking surgery by using original scoring with peritoneal cancer index and surgical complexity score, each. 

Author Response

I appreciate your comments. We evaluated the relationships between disease burden at the time of interval debulking surgery and surgical complexity and residual disease after surgery on survival outcomes in advanced-stage ovarian cancer patients treated with neoadjuvant chemotherapy. Our study showed that aggressive surgery at the time of interval debulking surgery dose not translate to survival benefit even in patient who achieved no residual disease after interval debulking surgery.

Reviewer 2 Report

Dear authors

The paper describes the effect of peritoneal cancer index (PCI), the complexity of surgery (surgical complexity score (SCS)) and residual disease in advance stage of ovarian cancer on PFS and OS.

The English is not always clear and for this reason the manuscript is hard to follow. This is a retrospective study in a small cohort of patients with a short follow up from a single center. The obtained data are in line with other published papers that are mentioned by the authors and this limit the novelty of the study. I suggest to submit the manuscript to a Journal that is more specialized in this field.

Author Response

Thank you for your comments. As the reviewer’s comments, there is a concern that the sample size is small to draw any conclusions. We tried to analyze the relationships between disease burden at the time of interval debulking surgery and surgical complexity and residual disease after surgery on survival outcomes in advanced-stage ovarian cancer patients treated with neoadjuvant chemotherapy. Our study showed that the patients who achieved no residual disease through aggressive surgery with high disease burden had poor than those who achieved no residual disease through less aggressive surgery with low disease burden. However, further larger-scale studies are necessary to validate the findings of the present study.

Reviewer 3 Report

    This is a well-written paper with clinical relevance. This manuscript evaluated the relationships between disease burden at the time ofinterval debulking surgery and surgical complexity score and residual disease after interval debulking surgery on survival outcomes in advanced-stage ovarian cancer patients. A high peritoneal cancer index at the time of interval debulking surgery intermediate/high surgical complexity score resulted in poor outcomes compared with those of patients with low peritoneal cancer index and low surgical complexity score. However, the R0 resection did not translate into survival benefit in patients with high peritoneal cancer index.
    I would like to suggest the authors to consider and comment on the effect of body composition changes during surgert and chemotherapy on survival outcomes in advanced-stage ovarian cancer. These patients often experience progressive muscle loss during treatment and patients with rapid muscle loss had worse surgical and survival outcomes (Rutten et al. J Cachexia Sarcopenia Muscle. 2016 Sep;7(4):458-66; Rutten et al. Eur J Surg Oncol. 2017 Apr;43(4):717-724; Huang et al. J Cachexia Sarcopenia Muscle. 2020 Jan 30. doi: 10.1002/jcsm.12524).

Author Response

Thank you for your comments. As the reviewer’s comments, we agree that severe loss of muscle during primary debulking surgery and chemotherapy might be associated with poor survival outcomes. Further studies are necessary to analyze the effect of progressive muscle loss on survival outcome in advanced-stage ovarian cancer patients treated with neoadjuvant chemotherapy. We analyzed the relationships between tumor burden and surgical complexity and residual disease after surgery on survival outcomes. Therefore the impact of sarcopenia on survival in ovarian cancer patients treated with neoadjuvant chemothearpy is beyond the scope of this article.

Round 2

Reviewer 2 Report

Dear authors

after revision, I don't see an improvement in the manuscript.

Author Response

Reviewer #2: Dear authors after revision, I don't see an improvement in the manuscript.

: As the reviewer’s comments, the entire manuscript has been revised. Editing of English language and style was performed again.